# The earliest evidence of high-elevation ice age occupation in Australia

Amy M. Way [1,2] ✉, Philip J. Piper [3], Rebecca Chalker[4], Dominic Wilkins[5], Erin Wilkins [5], Leanne Watson Redpath[5], Paul Glass[6], Marilyn Rose Carroll[7], Emily Nutman[8], Nina Kononenko[1,2], Michael Spate [9], Timothy T. Barrows[10], Duncan Wright[3] & Wayne Brennan[1,6]

Australia's Eastern Highlands have traditionally been viewed as a cold-climate barrier to Late Pleistocene (~35,000–11,700 years ago) mobility, with older evidence restricted to elevations below the periglacial zone. However, this model has not been adequately tested with regionally specific, high-resolution archaeological data. Here we report excavation results from a high-altitude (1,073 m) cave, Dargan Shelter, in the upper Blue Mountains, which indicate that occupation first occurred ~20,000 years ago, during the Last Glacial Maximum, making this the highest elevation Pleistocene site identified in Australia so far. Findings include multiple in situ hearths and 693 stone artefacts, several of which were sourced from sites along the mountain range, providing evidence for previously undetected interactions to the north and south and the repeated use of this cold-climate landscape during the Late Pleistocene. Our results align the Australian continent with global sequences, which indicate that cold climates were not necessarily natural barriers to human mobility and occupation.

High-altitude mountain landscapes pose severe challenges to human mobility. Globally, new deep-time cultural sequences are reconceptualizing these landscapes[1–3]. However, the timing of the earliest sustained occupation within periglacial environments in Australia remains unresolved. Previous archaeological work in Tasmania[4–6] and the Australian Alps[7–11] has only uncovered evidence for high-altitude occupation during the Holocene (11,700 years ago to present day). Occupation during the Late Pleistocene (~35,000–11,700 years ago) was restricted in all instances to at or below the periglacial limit (Fig. 1 and Supplementary Table 1).

In contrast, there is a growing body of evidence for high-altitude Pleistocene occupation in the Blue Mountains. Kings Table appears to provide the earliest evidence for high-altitude occupation in Australia;

however, both Bowdler[12] and Johnson[13] have questioned the validity of accepting the basal date of 22,300 ± 1900 radiocarbon years before present ([14]C yr BP) (SUA-158) as it comes from a fragment of charcoal found in association with a single small flake within a disturbed deposit. The next youngest date of 14,534 + 300 [14]C yr BP (SUA-194) is more secure and associated with 46 artefacts. This has, until now, provided the earliest accepted age for high-altitude occupation in Australia.

Here, we present results from Dargan Shelter in the upper Blue Mountains. This site produced a well-preserved, stratified archaeological sequence dated from the last glacial maximum (LGM) (19,000–21,000 calibrated years before present (cal. BP)[14]) to the later Holocene. Dargan provides evidence for human occupation of the Blue Mountains during the LGM, and the oldest evidence for high-altitude (>1,000 m

[1]Discipline of Archaeology, The University of Sydney, Sydney, New South Wales, Australia. [2]Australian Museum Research Institute, Australian Museum, Sydney, New South Wales, Australia. [3]School of Archaeology and Anthropology, The Australian National University, Canberra, Australian Capital Territory, Australia. [4]Dharawal Environment and Heritage, Tahmoor, New South Wales, Australia. [5]Dharug Custodian Aboriginal Corporation, Windsor, New South Wales, Australia. [6]Blue Mountains Aboriginal Community, Wentworth Falls, New South Wales, Australia. [7]Corroboree Aboriginal Corporation, Rouse Hill, New South Wales, Australia. [8]Department of Archaeology and Natural History, The Australian National University, Canberra, Australian Capital Territory, Australia. [9]Department of Archaeology and History, La Trobe University, Bundoora, Victoria, Australia. [10]Chronos Radiocarbon Laboratory, University of New South Wales Sydney, Kensington, New South Wales, Australia. ✉e-mail: amy.way@sydney.edu.au

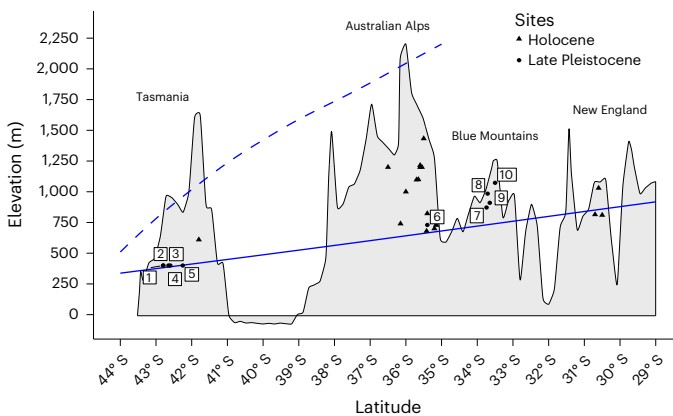

**Fig. 1 | Holocene and Late Pleistocene sites in eastern Australia with an elevation at or above the periglacial limit.** The elevation from Tasmania to New England is indicated by a thin black line and the shaded area beneath it. The solid blue line indicates the approximate lower limit of the periglacial zone, and the dashed blue line indicates the equilibrium line altitude for glaciers during the LGM. Sites are plotted by latitude (S) and elevation in metres above sea level (Supplementary Fig. 1 and Supplementary Table 1). Holocene sites are marked with a black triangle. Late Pleistocene sites (black circle) are numerically labelled: (1) Bone Cave, (2) Stone Cave, (3) Nunamira Cave, (4) Tiata Mara Kominya, (5) ORS 7, (6) Birrigai, (7) Kings Table, (8) Lyrebird Dell, (9) Wall's Cave and (10) Dargan Shelter. Glacial and periglacial limit data are from refs. 19 and 47.

above sea level) occupation in Australia. This evidence requires a paradigm shift in our understanding of Australia's mountains[15] as a barrier to human mobility during the LGM[16–18].

## Results

### Site, stratigraphy and dating

Dargan is a large rock shelter in Dharug Country near Lithgow, NSW (Fig. 2), below the modern subalpine zone in the sclerophyll forest. It sits on the southern side of a spur leading off the Bell's Line of Road ridgeline, which is one of two main pathways through the Blue Mountains. While there is no definitive way of identifying which groups accessed the mountains in the deep past, multiple groups were probably connected to this country. The cave probably presented an important focal point for people positioned within networks extending between the western interior, the Cumberland coastal plains and the country to both the north and south. Today, the Wiradjuri, Gomeroi, Darkinjung, Dharawal, Wonnarua, Gundungara and other groups hold traditional connections to this region. The cave has a small amount of faded rock art, including a child-sized hand stencil and two forearm stencils. D-stretch photography indicated that a number of other motifs existed in earlier times but are no longer clearly visible. Today it is considered by local custodians to represent a family space of high cultural importance.

Archaeological deposits were only identified within the northern quarter of Dargan (Fig. 2d). Here, deposits were excavated to bedrock at a maximum depth of 2.3 m. This produced a well-defined stratigraphic sequence that preserved cultural material from 5–10 cm to 1.6 m depth below the surface. The stratigraphy, super-positioning of dated samples without a single inversion, artefact refits and age–depth model indicate that the rock shelter sediments are not turbated. The modelled radiocarbon ages suggest that the lowest artefacts were plotted adjacent to a hearth with an age of 21,700–19,990 cal. BP (see model A in Fig. 3, and Supplementary Tables 2 and 3) providing the oldest evidence of occupation above 700 m in Australia, and evidence for occupation well above the tree line during the LGM. Artefact density peaked at 18,000–16,000 cal. BP, showing repeated use of this site during this interval.

Figure 4 and Supplementary Figs. 2–5 summarize the stratigraphic correlation between the four excavated trenches and the relationship

between sedimentary layers, excavated units, ages and artefacts. The sequence for which age–depth modelling of the dating results offers a robust chronology (Fig. 3, Supplementary Fig. 6 and Supplementary Table 2) can be summarized as follows.

The lowermost stratigraphic layers (16–18) did not hold any artefacts or concentrations of charcoal or ash. They consisted of sterile light-pinkish brown slightly gravelly, silty, coarse sand (Supplementary Fig. 7), which contained increasingly larger angular fragments of sandstone with depth onto the bedrock. These layers sloped at a ~15° inclination from the rear cave wall towards the entrance, but the upper surfaces appear to have become more horizontal as they reached pits 10E to 7E. An isolated charcoal fragment located in layer 18 between roof fall fragments provided a possible basal radiocarbon age of at least 50,000 [14]C yr BP (S-ANU74932) for the blocking of the cave entrance. This is not interpreted as evidence of human presence at this time.

These deposits were overlain by several light reddish-brown to pink poorly sorted gravelly, silty, coarse sands (layers 11–15). These contained numerous artefacts and concentrations of ash, charcoal and burnt sediments (localized reddening of the sand) and heat damaged artefacts that indicate the presence of in situ hearths. These layers date from the LGM, from 21,700–20,000 cal. BP (Start Dargan) to 16,200–15,590 cal. BP (Supplementary Fig. 2, see also Supplementary Tables 2 and 3). During this time periglacial conditions extended to the upper reaches of the Blue Mountains above ~600 m and temperatures would have been at least 8.2 °C cooler than today and vegetation would have been sparse[19–21]. Pollen analyses, although with low counts, indicate no arboreal pollen at the site (see Supplementary Table 4) and regional pollen records indicate the treeline was ~400 m below the site[22] (see also the Supplementary Information). Little firewood would have been available locally, and water sources would have been frozen through winter.

The final stages of occupation at Dargan during the LGM were represented by two slightly pinker layers of coarse sands (layers 9 and 10) similar to layers 11–15, which date from ~16,000 to 13,000 cal. BP at outer limits. These represent the period of transition from open, herb dominated landscapes to arboreal Myrtaceae and Casuarinaceae dominated vegetation reflecting the initiation of wetter and warmer conditions from the end of the LGM[21,23]. This supports the nearby record from Mountain Lagoon of a transition from sedgeland to woodland/forest[21]. The pollen spectrum from this time onwards demonstrates little vegetation variability[24,25] with firewood readily available from the onset of the Holocene. The oldest Holocene layers preserved evidence for intensive hearth construction, reflective of this increased availability of timber, which produced thick charcoal rich and reddened sand layers. The lowest of the Holocene layers, layer 8, consisted of a dark reddish-brown gravelly, silty, coarse sand laden with charcoal and ash containing numerous stone artefacts dating from 12,120–9,670 cal. BP to 8,990–6,510 cal. BP (Fig. 3 and Supplementary Table 2). This was succeeded by a sequence of four layers (4–7) of pinkish to reddish brown gravelly, silty coarse sand dating to between ~6,000 and 1,500 cal. BP. Layers 6 and 7 contained discrete lenses of charcoal and ash suggesting multiple periods of site occupation and use within a broader phase of sediment deposition. The most recent in situ archaeological deposit was a brown gravelly, silty, coarse sand containing high concentrations of charcoal and ash just below modern ground surface (layer 3). This latest recognizable phase of habitation dates to 500–330 cal. BP (S-ANU72611).

### Artefacts

A total of 693 stone artefacts were excavated across the three seasons, of which 680 were analysed. Artefacts were found in stratigraphic layers 2–15, with 117 from stratigraphic layers older than 16,000 cal. BP (layers 11–15) (Supplementary Table 5). Figure 4 shows the artefact density by stratigraphic layer and age. The four lowest artefacts (layers 14 and 15) were deposited during the LGM. This initial visitation involved the

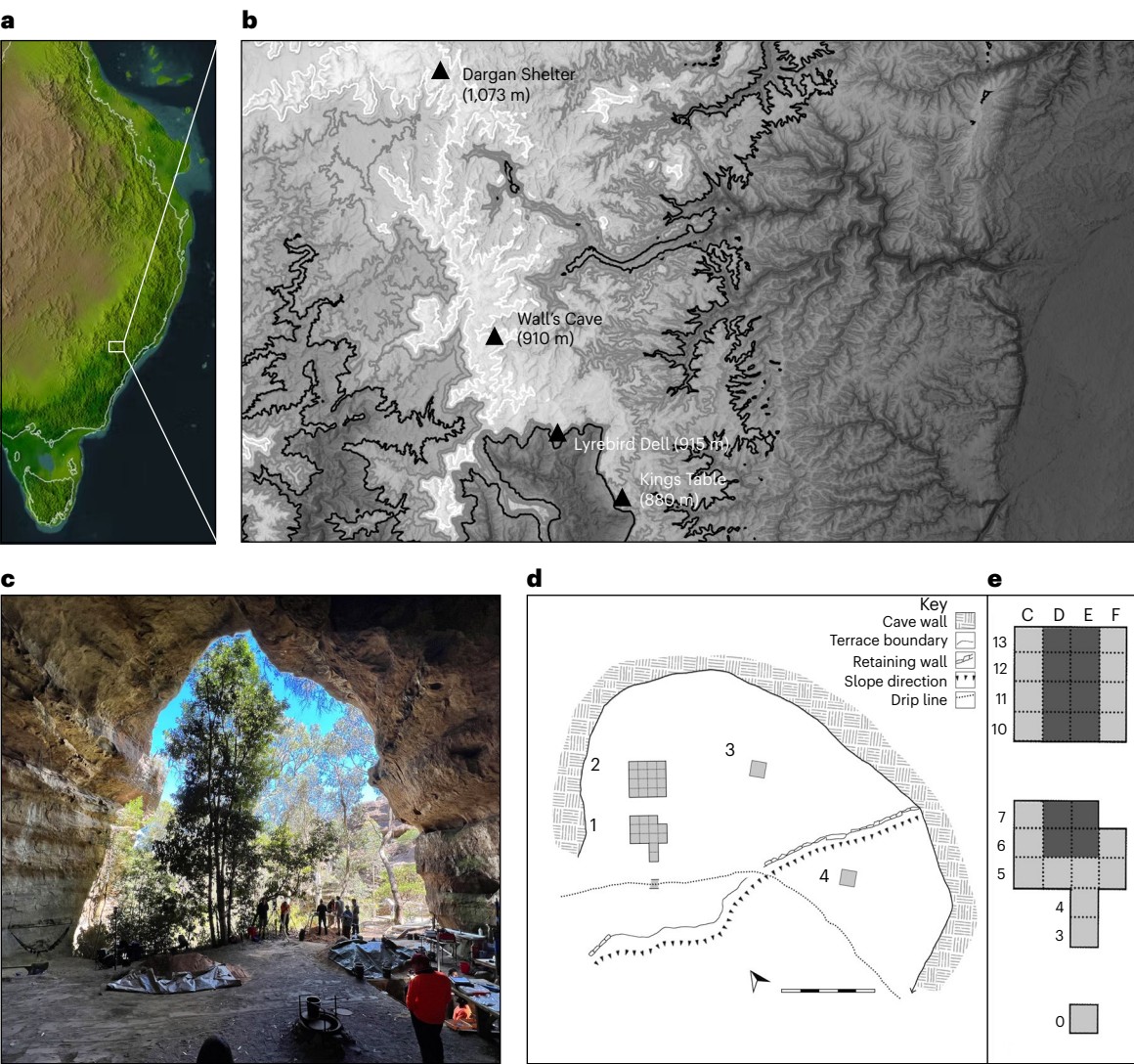

**Fig. 2 | Dargan Shelter—location, elevation and plan of excavation. a**, The location of Dargan Shelter in eastern Sahul during the LGM. **b**, The elevation of Dargan Shelter and other Late Pleistocene sites in the Blue Mountains in relation to the periglacial limit during the LGM (~600 m). **c**, Photo of Dargan Shelter taken from the rear of the shelter and looking out to the west. **d**, A plan of the excavation showing the location of the four trenches. **e**, A plan of the excavation of trenches 1 and 2 and the dripline test pit (0). Note that the 12 darker squares were excavated to basal sterility or bedrock, which was reached at a depth of ~2 m. The lighter squares were excavated to ~1 m. Panel **a** courtesy of Damian O'Grady and Michael Bird. In panel **b**, map data © Commonwealth of Australia (Geoscience Australia) 2021, CC BY 4.0; map created with ArcGIS software by Esri.

discard of a small number of flaked artefacts, which had been manufactured elsewhere. The lowest complete flake was found in layer 15. It was made from local claystone, showed use-wear on the distal end, measured 47 mm in length and weighed 6.9 g (Supplementary Fig. 8, artefact no. H-17). Artefact numbers steadily increased during the LGM. Refitted artefacts were present at ~16,000 cal. BP, providing evidence for on-site stone working in a periglacial environment.

Raw material selection, maximum flake length and retouch rates remained relatively stable through time, suggesting no major techno-typological change, despite considerable palaeo-climatic shifts (Supplementary Fig. 9). The percentage of retouched flakes shows non-significant variation from 4.3% in the Pleistocene to 4.1% in the Holocene. Complete flakes decrease slightly over time in size from an average length of 16 mm and weight of 3 g in the Pleistocene to 14.7 mm and 1.3 g in the Holocene. The use of the dominant raw materials of quartz and claystone, which are both available locally as creek cobbles and bedded veins, also only shifts marginally over time (Supplementary Fig. 9a).

Most of the claystone lithics from Dargan Shelter were made from the locally available Burragorang Claystone Member; however, ten fine-grained cryptocrystalline flakes were identified as chemically distinct and are considered exotic, with the closest chemical match ~50 km to the south-west at Jenolan (Supplementary Table 6 and Supplementary Fig. 10). Four of these 'unidentified silicious' flakes were found in layers 11 and 12, which date to 16–17,000 cal. BP, suggesting a southern entry to the region during the LGM. In addition, three ferruginous black quartzite artefacts, including a hammerstone, were matched to material in the Hunter region ~150 km to the north (Supplementary Fig. 10). These were from Pleistocene layers 10 (~15,000 cal. BP) and 13 (~18,500 cal. BP) and indicate that people were coming in from or connected to the north, as well as the southern ranges during the LGM.

A notable Late Pleistocene artefact was a sandstone grinding slab dating to 13,460–13,190 cal. BP (S-ANU 74936) (Supplementary Figs. 11 and 12). It had two linear grooves characterized by distinctive abrasive wear consistent with shaping by abrasion of bone or wooden artefacts such as needles, awls, bone points and nose points. A notable

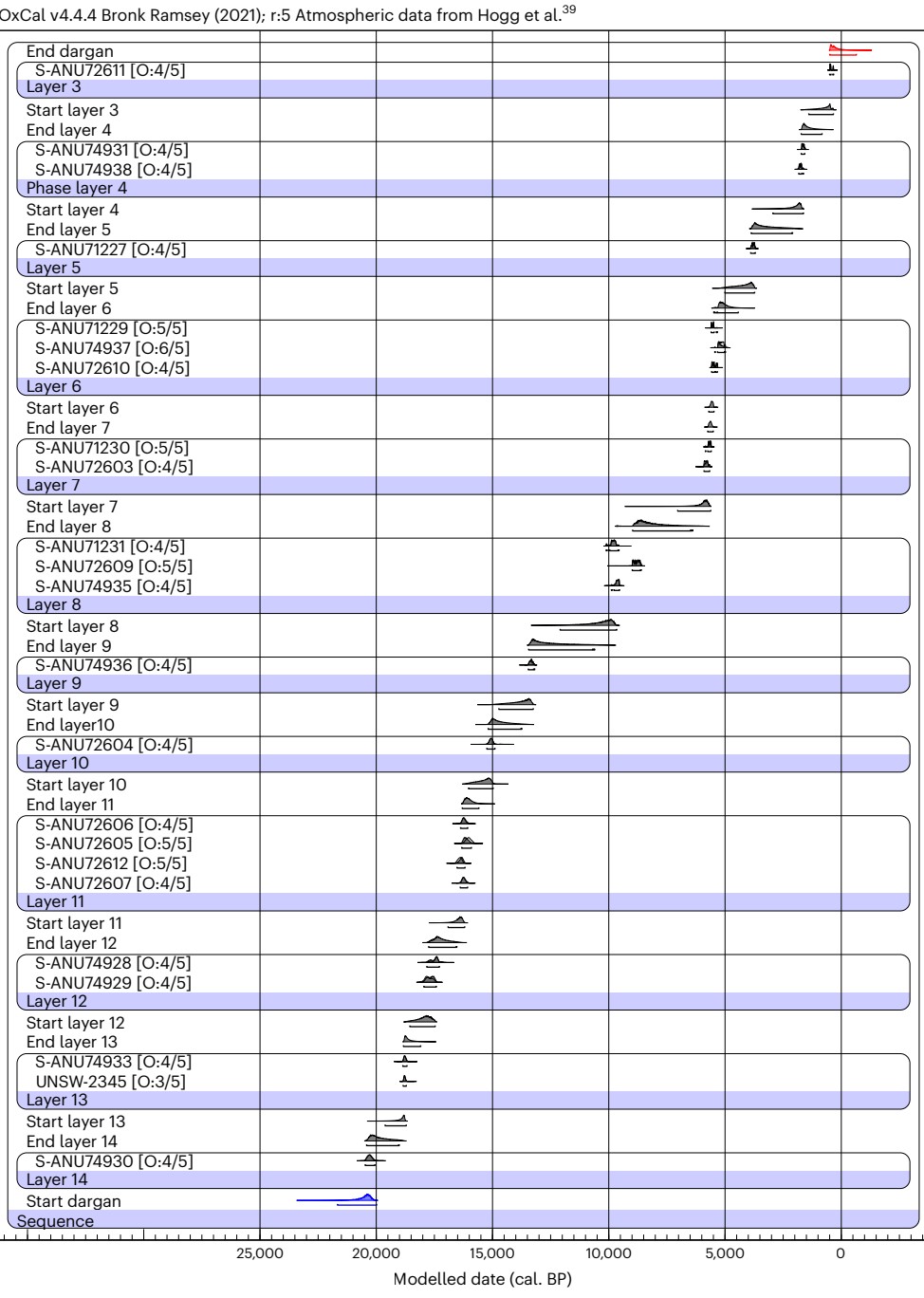

OxCal v4.4.4 Bronk Ramsey (2021); r:5 Atmospheric data from Hogg et al.[39]

**Fig. 3 | General T-type model (model A).** The layers are organized in stratigraphic order with the oldest at the bottom. Figure created with OxCal v4.4.4.

Early Holocene find was a basalt anvil in layer 8 (8,990–8,610 cal. BP). It consisted of a split river pebble identified as hornfels with distinctive sub-circular patch and impact marks consistent with cracking hard woody nut and seed shells (Supplementary Fig. 13).

## Discussion

The excavation of Dargan Shelter provides evidence for use of high-altitude Australian mountains (>1,000 m) during the LGM. Artefacts and hearths dated to between 20,000 and 16,000 cal. BP show that alpine mountain occupation was more persistent than previously known and set an evidential framework for understanding high-altitude Late Pleistocene occupation.

Periglacial conditions during this time included an absence of trees, dramatic cooling and seasonal freezing of water sources. A grassland tundra probably supported ephemeral mammal and bird populations during summer. From east to west the mountain range is approximately 100 km, requiring a minimum crossing time of 5 days. However, the pXRF analysis provides evidence for Late Pleistocene connections to the southern and northern ranges, suggesting that this region was not simply a least-cost pathway from east to west, but rather may have been a stopping point for groups spending substantial time travelling along the mountain range during the warmer seasons. People may have been utilizing this site as a stopover point while travelling to undertake ceremony[26,27] or to access resources unable to be obtained at lower elevations[28].

The peaks in artefact discard and hearth construction seen during the LGM and the beginning of the Holocene closely parallel the pattern seen in the oldest site to the east, PT 12, in the Sydney Basin[29]. This

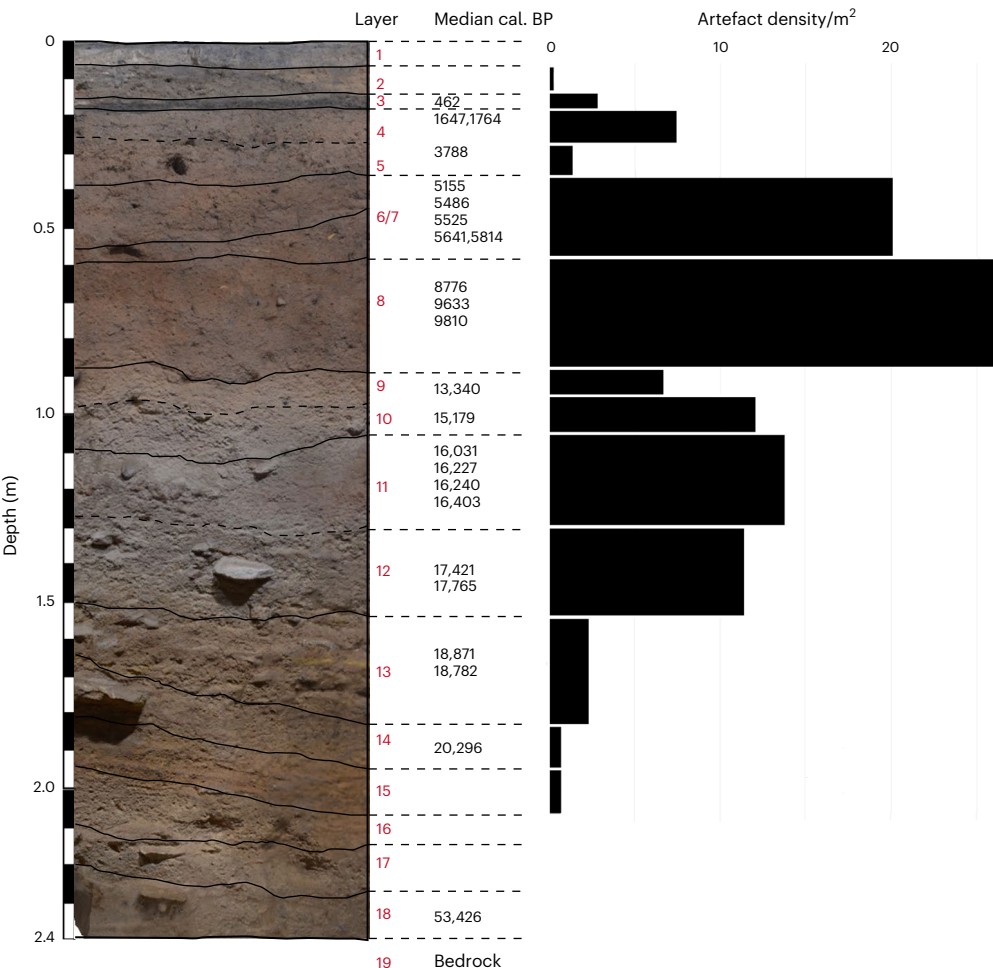

**Fig. 4 | Artefact densities in each stratigraphic layer.** The median ages are based on calibrated unmodelled dates (Supplementary Table 3). Stratigraphic layers 6 and 7 have been combined owing to their overlap in ages.

coastal plain site shows activity commencing before 36,000 cal. BP with regular repeated use during the LGM. Results from Dargan Shelter demonstrate that connections between the Sydney Basin and the Blue Mountains, which were clearly in place by the terminal Pleistocene[17], may have begun much earlier.

The Dargan results are also consistent with recent cross-continental evidence for early occupation of high-altitude habitats across Europe[30], Africa[31], the Americas[32] and Asia[3]. This evidence from Australia demonstrates that the occupation of mountain ranges >1,000 m in a periglacial landscape was also found on this continent.

## Conclusions

Dargan is the earliest site to be excavated in high-altitude Australia. The excavation revealed evidence of a continuous sequence of occupation from the LGM ~20,000 until ~400 cal. BP, refuting the hypothesis that people first entered altitudes above the periglacial limit in Australia only several millennia after the LGM[16–18].

While this does not represent the initial peopling of south-eastern Australia, the evidence here provides demonstration of repeated human movement through and adaption to a periglacial environment during the LGM and supports demographic modelling, which postulates pathways through environments previously conceived as uninhabited[33].

## Methods

The archaeological excavation was completed under Aboriginal Heritage Impact Permit 4,770, issued to the Australian Museum and Australian National University on 17 June 2021 pursuant to s.90 of the National Parks and Wildlife Act 1974, which governs the investigation of Aboriginal objects and places in NSW. The project was initiated in 2019 to bring archaeologists and local custodians together to assess the cultural importance of previously un-excavated rock shelters on the Bell's Line of Road in the Blue Mountains. Consultation followed the Office of Environment and Heritage's Aboriginal cultural heritage consultation requirements (Department of Climate Change, Energy, the Environment and Water 2010) with 11 groups or individuals registering an interest. A project specific Blue Mountains Indigenous Advisory Committee was then formed, with Committee members partnering in the design, implementation and reporting of project results.

The excavation and laboratory analyses were undertaken with the Indigenous Advisory Committee's fully informed approval. All aspects of the project adhere to the Australian Museum's Indigenous Cultural and Intellectual Property protocols. After completion, the artefacts will be returned to country, with Advisory Committee members deciding how they will be reburied or displayed.

## Archaeological excavation

Dargan Shelter was excavated over three seasons from April 2022 to March 2023. The team excavated 7.5 m² of deposit to a maximum depth of 2.3 m (Fig. 2, trenches 1 and 2) revealing evidence of a sequence of natural and cultural deposits at the north end of the cave that had accumulated from before the LGM until the sub-recent. trenches were gridded into 50 cm × 50 cm squares and single excavation units did not exceed 5 cm in depth. Careful hand excavation was employed with

features and samples plotted in three dimensions. All sediment was dry sieved through a 3 mm screen. Lithics recovered from the screen were provenanced to the 50 cm × 50 cm × 5 cm unit.

There was a distinct boundary between the lowest Holocene layer (8) and uppermost Late Pleistocene layers (layers 9 and 10), and between the top of the in-situ deposit (layer 3) and the overlying sand distributed during the 1980s (layer 2). The clear distinction between these layers allowed them to be excavated as single contexts. To ensure no contamination between layers 8 and 9/10, a thin (1 cm) clean-up unit was excavated after the excavation of layer 8 and before the commencement of layer 9. This yielded three stone artefacts, which were excluded from the analysis. The excavation of layer 2 included the removal of the contact interface to ensure that layer 3 did not contain any mixed material. There were no discernible interfaces within epochs, but rather a gradual sedimentary build up over time, as evidenced in the sand grain size distribution (Supplementary Fig. 7) and dates (Fig. 3).

Season 1 (April 2022) involved the excavation of five 50 × 50 cm test pits (Fig. 2, E0, E3, E4, E5 and E6) to a depth of 60–70 cm. The test-pits were positioned from the dripline to the interior of the cave. Seven artefacts were found in the two innermost test pits within interleaved layers of burnt sand and charcoal hearths, indicating that cultural deposits were preserved inside the shelter.

The two artefact yielding test pits E5 and E6 were expanded in season 2 to form trench 1, which measured 2 m × 1.5 m, and was excavated to a depth of 2 m. Trench 2 was also opened. It measured 2 m × 1 m and was excavated to a depth of 1.1 m. Two exploratory 1 m × 1 m trenches (3 and 4) were also added to the central interior and the lower terrace of the cave to investigate spatial variation (Fig. 2d).

Trenches 3 and 4 revealed sloping bedrock at depths of ~1 m and ~50 cm, and all deposits except for the modern surface sequence were completely sterile (Supplementary Fig. 3). Sand redistributed by the landholder in the 1980s presented as the upper layers of trenches 3 and 4 and overlay the dark humic, charcoal and ash silty sand deposits of the original ground surface as seen in trenches 1 and 2 (Supplementary Fig. 2). Beneath this was a sequence of two or three yellowish brown gravelly, silty sand deposits containing increasingly large angular fragments of sandstone with depth. These layers overlay densely packed, large fragments of sandstone roof fall, some exceeding 0.5 m in length with a thickness of ~0.1 m, which sloped at an angle of as much as ~25° towards the cave entrance (Supplementary Fig. 3). All these deposits were very similar in sediment composition to the culturally sterile layers recorded towards the base of the main excavation area to the north (layers 16–18, Supplementary Fig. 2). Trenches (3 and 4) indicated that none of the cultural deposits recorded at the north end of Dargan Shelter in trenches 1 and 2 extended more than 8 m from the north wall.

In season 3 (March 2023), trench 2 was expanded to an area of 2 m² with the central 1 m × 2 m excavated to bedrock, which was reached at a maximum depth of 2.3 m (Supplementary Fig. 2). The basal deposit (layer 18) in trench 2 consisted of closely packed very large angular and inclined ~20° fragments of sandstone roof fall some of which spanned the breadth of the pit. A ground-penetrating radar survey confirmed that dense roof fall extended as far as the cave entrance with approximately the same inclination. At some point before the LGM, this part of the entrance had become blocked, possibly by large pieces of cave overhang collapsing, which inhibited the downslope movement of sediments from the northern interior of the cave. This resulted in the accumulation of more than 2 m of natural and cultural deposits at the north end of Dargan Shelter. Continuing roof fall and sediment dropping from the dripline added to this development and is represented in the archaeological record by deposit 20 in pit OE (Supplementary Fig. 2).

### Charcoal sample collection
Charcoal samples were collected in situ during excavation, plotted in three dimensions, drawn on recording sheets and photographed. Supplementary Table 3 lists dated charcoal samples, including eight failed

samples, and feature information. Hearth charcoal was differentiated from potential cultural burning/wildfire charcoal by its consolidation in bounded concentrations and association with burnt sediment and/or ash and/or heat damaged artefacts (Supplementary Fig. 14). Fragmented pieces outside of these concentrations were classified as non-hearth charcoal. As timber was not present in this landscape during the LGM, charcoal in layers 9–15 was probably introduced to the site through anthropogenic fires. Charcoal was much more common in the upper Holocene layers than in the lower Late Pleistocene deposits, and this was reflected in the numbers of samples that were collected (Supplementary Table 3). High levels of roof fall throughout the deposit, increasing in frequency and size with depth, limited the opportunity for vertical migration of all samples (Supplementary Fig. 14).

### Radiocarbon dating
In total, 32 potential samples were submitted to the Australian National University and UNSW radiocarbon laboratories for dating. Of these, eight samples were considered unsuitable for dating as they consisted of small black fragments that were not charcoal (Supplementary Table 3). One of these samples was from Holocene layer 8, and seven samples were from the basal Pleistocene layers 15–18. This left just one suitable sample from layer 14 for dating.

The 24 charcoal samples that passed initial quality control were cleaned with an acid–base–acid pre-treatment protocol at the Australian National University[34]. After cleaning, sediment and visibly degraded material were removed with a scalpel and charcoal samples were then submitted to a series of acid (1 M of HCl, 30 min, 70 °C), base (1 M of NaOH, 1 h, 70 °C, repeated until solution colourless) and acid (1 M of HCl, 30 min, 70 °C) washes with each step followed by rinsing with ultrapure water. Cleaned samples were combusted in a sealed quartz tube in the presence of CuO wire and Ag foil. The resulting $CO_2$ was purified and collected cryogenically before graphitization with $H_2$ over an Fe catalyst and measurement at the Australian National University[35]. Dates were calculated following[36] using a $\delta^{13}C$ value measured by AMS. One sample (UNSW-2345) was dated at the Chronos Radiocarbon Laboratory at the University of New South Wales[37].

### Radiocarbon modelling
Dates have been calibrated in OxCal 4.4 (ref. 38) with the Southern Hemisphere SHCal20 (ref. 39) calibration curve. All calibrated uncertainties are given at 2 sigma or 95.4% probability throughout except for Supplementary Table 1 where they are presented at 1 sigma (68.3%), per radiocarbon conventions. Both the Bayesian models applied here in OxCal place radiocarbon dates within phases organised into a single sequence (Fig. 3, Supplementary Fig. 6 and Supplementary Table 2).

The two models (outlier models) are designed to test dataset sensitivity to the old wood effect in charcoal. The general T-type outlier model (model A[40]) assigns all samples a 5% prior probability of being an outlier (following the Student's *t* distribution, with five degrees of freedom). Model flexibility allows samples to be both too old or too young. The modified charcoal or charcoal plus model (model B[41]) predicts that charcoal samples are probably accurate, but if inaccurate, are most likely to be too old owing to the old wood effect. However, the model also permits a small proportion of samples to be too young to account for possible, though limited, post-depositional movement between stratigraphic contexts. All charcoal samples are assigned 100% prior probability of being an outlier. The sequences presented here have 23 dates (S-ANU74932 is excluded) that were associated with human activity at Dargan Shelter (Fig. 3, Supplementary Fig. 6 and Supplementary Tables 2 and 3).

The calibrated unmodelled and modelled radiocarbon dates using the general T-type (model A) and charcoal plus outlier (model B) models demonstrate good agreement between stratigraphic context and age. We have chosen to illustrate the general T-type model with posterior outlier values in the main text (Fig. 3, model A) and the charcoal plus

outlier model (Supplementary Fig. 6, model B) in the Supplementary Information. The samples in model A (Fig. 3) show that all but sample S-ANU74937 (with a 6% value) have posterior outlier probability values of 5% or less. This has two important implications. Firstly, it illustrates that none of the dates demonstrate an 'old wood' effect, and/or are too young or too old for their respective locations within the chronological sequence. Secondly, it indicates that there is not a single age inversion in the stratigraphic sequence. This illustrates the integrity of the archaeological stratigraphy and the quality of sample selection. Both models indicate that initial occupation at Dargan Shelter was probably sometime between 22,000 cal. BP and 19,900 cal. BP (Supplementary Table 2).

### Sediment analysis

Sediment samples were taken at 5–10 cm intervals in trench 2, Sq. 10E using 3 cm ø tubes. Around 0.5 cm³ of sediment was subsampled in the laboratory and treated with 10 ml 10% HCl to remove carbonates. Organic materials were oxidized with 40% $H_2O_2$, then samples were mechanically agitated in 30 ml of 10% sodium hexametaphosphate for a minimum of 2 h. Particle size readings were taken using a Malvern 2000 Mastersizer with Hydro2000Mu dispersion attachment. Output data were analysed and described using the Gradistat v.9 macro for Microsoft Excel[42] following the geometric method[43].

Sand was the dominant sediment size fraction through the full depth of trench 2 (Supplementary Fig. 7), ranging from 59% to 89% relative abundance (mean 79%). Silt-sized sediment ranged from 8% to 32% (mean 16%) and clay sized clasts ranged from 1.7% to 9% (mean 4.1%). Higher relative abundances of silt and clay-sized sediment appear correlated with higher concentrations of artefacts between layers 4 and 12 (Supplementary Fig. 7). These layers also have mean particle size values below the long-term mean (273 µm).

### Pollen analysis

Four sediment samples were passed through a 1-mm sieve then 1 cc was subsampled. Samples were spiked with a known quantity of *Lycopodium* marker spores (20,408 ± 543; Lund University batch 101023-231) and dispersed in 10% sodium hexametaphosphate. Pollen extraction followed standard heavy liquid procedures. Samples were treated with 10% NaOH and 10% HCl to remove humic acids and carbonates. Samples were separated with a sodium polytungstate heavy liquid at specific gravity 1.95. The light fraction was then treated with an acetolysis mixture of 9:1 acetic anhydride and sulfuric acid. Washed samples were then mounted for pollen counting.

### Artefact analysis

No faunal remains or artefacts made from organic materials were recovered owing to the high acidity of the deposits. Overall, 693 stone artefacts were excavated across the three seasons, of which 680 were recorded in three dimensions. The 13 excluded artefacts were collected during clean-ups. Mass (measured to 0.01 g) and raw material (5× hand lens) were recorded for every artefact. Every artefact >3 mm was catalogued as either a flake, retouched flake, core, flaked piece or non-diagnostic angular fragment (Supplementary Table 5). Following[44] flakes had one positive ventral face, or part thereof; retouched flakes had negative scars removed after the ventral face was created; cores had one or more negative scars but no positive scars; and flaked pieces had one or more negative scars, but some ambiguity regarding a ventral face. The angular fragments did not have either a clear negative or positive scar and were often heat damaged. Heat damage, consisting of crenation, potlids or colour alteration was also recorded for all non-quartz artefacts. Flakes were further classified by breakage (complete, longitudinally cone split, marginal, margin missing, proximal, medial and distal). Flake dimensions (digital callipers to 0.01 mm) and attributes were recorded for each flake. Maximum length was measured from the point of impact to the distal margin on the ventral

surface in the direction of percussion. Maximum width was measured at the widest point perpendicular to maximum length and maximum thickness at the thickest point between the dorsal and ventral surfaces. Platform width was measured from the junction of the ventral surface and the platform. Platform thickness was measured perpendicular to platform width at the point of impact. Terminations (feather, step, hinge or plunging) were recorded as well as an estimation of dorsal cortex (0 for 0%, 1 for 1–25%, 2 for 26–75%, 3 for 76–99% and 4 for 100%). Platforms were recorded as single scar, multiple scars, facetted, cortical, linear or shattered.

Core dimensions included maximum length, width and thickness and with core length defined as the longest axis. Width was measured perpendicular the length and thickness at the junction of width and length. Cortex was estimated as per flakes. Cores were classified as bipolar, multi-platform, and single platform. Refitting was performed to examine reduction sequences and artefact movement.

### pXRF analysis

In total, 109 non-quartz lithics from the Dargan Shelter were geochemically characterized with a Bruker Tracer 5 g handheld portable X-ray fluorescence machine (serial number 900G10419). Samples were analysed three times on two different settings to achieve good chemical resolution on both the major rock forming oxides and the trace element composition. The first set of assays targeted light elements and major oxides, with the machine using no filter and being set to 15 keV and 45 µA and running for 120 s. The second set of assays targeted trace elements using the Cu75, Ti25, Al200 'green' filter, with a particular focus on Ti and on Rb. Sr, Y, Zr and Nb. These assays ran for 60 s at 50 keV and 35 µA.

The reference materials ANU2000 (ANU in-house obsidian) and BHVO-1 (international Hawaiian basalt chemical standard) were run to check for drift after every 20 artefacts, and the light element spectra was normalized to the Rh L peak of ANU2000 in Artax (Bruker). Trace element spectra was normalized to the Rh K alpha of BHVO-1 through the same procedure. Deconvolution was performed in Artax, and the regression lines fitted in Microsoft Excel. These regression lines were built with 47 international and in-house standards. Preferred values from GeoReM were used for the CRMs[45]. Eighteen of the Australian National University and Auckland University inhouse standards were made with WDXRF. ANU2000 used LA-ICPMS certified values. A 100% silica blank was included, as was CRM 45 d from OREAS, which used compiled values from multiple laboratories (see Supplementary Table 7 for full list of standards). Spectra from these standards was collected and processed using the methods described above. No filter calibration curves were used for all major oxides except TiO2. Cu75:Ti25:Al200 'green' filter was used for TiO2, Rb, Sr, Y, Zr and Nb.

### Use-wear analysis

Use-wear analysis was carried out on the sandstone slab from layer 9 (~13,200 cal. BP) and basalt anvil from layer 8 (~8,500 cal. BP) with a Dino-LiteTM (AM413ZT) digital microscope using magnifications from ×10 to ×50, with direct vertical light combined with an additional oblique light from an external source, and with the metallurgical microscope Olympus BX60M fitted with vertical incident and transmitted light and providing magnifications from ×50 to ×1,000. The images of use-wear traces and residues were taken with the Olympus DP72 camera and Soft Imaging System GmbH attached to the metallographic microscope.

The surface of the sandstone slab was initially scanned and residue was extracted following a protocol described in detail in ref. 46. Residues were sampled from worn surfaces within the grooves. Samples were extracted by applying up to 40 µl of distilled water and then acquiring the sample with an adjustable pipette fitted with disposable nylon pipette tips. The residue solution (~10 µl) was mounted on a clean glass slide. A clean cover slip was placed over the residue

mixture and sealed in four corners with clear nail varnish. Residue samples were examined under transmitted light using an Olympus B×60 M microscope. The residues included recent fibre and particles of charcoal that are associated with contaminations and not related to the use.

After residue extraction, the surface on the slab was microscopically examined. The slab displayed heavy weathering of the outer surface of the rock and a harder inner layer (Supplementary Fig. 12). Because of the loosely cemented and easily dislodged sand grains of the weathered outer surface, the slab was cleaned for use-wear analysis by very light washing under running water.

## Data availability

The authors confirm that all data generated or analysed during this study are included in this published article. Additional primary data on excavation depths are described in Supplementary Figs. 2–5 and data on the stone artefacts in Supplementary Table 5. The primary data on the radiocarbon dates are available in Fig. 3, Supplementary Fig. 6 and Supplementary Tables 2 and 3. The methods of dating and choice of Bayesian models are described in the Methods. The codes for the general T-type and charcoal plus outlier models applied are included in the Supplementary Information. The artefacts described in this paper are currently stored in the Archaeology Hub at the University of Sydney and will be returned to the Dharug Custodians upon completion of the study.

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

## Acknowledgements

We thank the members of the fieldwork team, K. Chalker, L. D. Carroll, T. Pal, W. Logue, E. Trewlynn and R. Lawson, as well as students on the University of Sydney and Australian National University Blue Mountains field schools, S. Mabon, L. Roach, M. Y. Choi, L. Jones, A. Mavin, A. Serena, A. Collins, N. Crockford, I. Williams, J. Scholefield, N. Taylor and J. Vukovic as well as M. Warwick, E. Harris and D. Johnston for their assistance in the field. We thank P. Byrne for conducting the GPR survey, M. Robinson for photographing the artefacts, R. Hawkins for assistance with the lithic refitting program, R. Pogson for facilitating access to the Australian Museum's geological collections, P. Vaiglova for advice on Bayesian analyses and M. Corrigan for facilitating access to the site. Funding was provided by the Australian Museum Foundation (grant no. 1495 to A.W.). The funders had no role in study design, data collection and analysis, decision to publish or preparation of the manuscript.

## Author contributions

This manuscript was authored in partnership with the Blue Mountains Indigenous Advisory Committee. A.W., W.B., D.W., E.W. and L.W.-R. conceptualized the study. A.W., P.P., E.N., N.K., M.S. and T.B. carried out the methodology. A.W., W.B., P.P., R.C., D.W., P.G., E.N., N.K., M.S., D.W. and M.C. carried out the investigation. A.W., P.P., M.S., E.N. and T.B. performed vizualization. A.W. carried out funding acquisition and project administration. A.W., W.B., R.C. and P.P. supervised the study. A.W., P.P., E.N., N.K., M.S. and T.B. carried out writing—original draft. A.W., P.P., W.B., E.N., N.K., M.S., T.B., D.W., R.C., D.W., E.W., L.W.-R. and P.G. carried out writing—review and editing.

## Funding

## Competing interests

The authors declare no competing interests.

## Additional information

**Correspondence and requests for materials** should be addressed to Amy M. Way.

