## [Peer Review File · Nature Human Behaviour]

The earliest evidence of high-elevation ice age occupation in Australia

Corresponding Author: Dr Amy Way

Version 0:

Decision Letter:

10th September 2024

Dear Dr Way,

Thank you once again for your manuscript, entitled "The earliest evidence of high-elevation ice age occupation in Australia," and for your patience during the peer review process.

Your manuscript has now been evaluated by 2 reviewers, whose comments are included at the end of this letter. Although the reviewers find your work to be of interest, they also raise some important concerns. We are interested in the possibility of publishing your study in *Nature Human Behaviour*, but would like to consider your response to these concerns in the form of a revised manuscript before we make a decision on publication.

To guide the scope of the revisions, the editors discuss the referee reports in detail within the team, including with the chief editor, with a view to (1) identifying key priorities that should be addressed in revision and (2) overruling referee requests that are deemed beyond the scope of the current study. In the case of your manuscript, we are particularly concerned that both of our reviewers raise issues with your reporting and presentation of data, as well as your discussion of results. Please ensure these concerns are addressed in full in your revised manuscript.

In sum, we invite you to revise your manuscript taking into account all reviewer and editor comments. We are committed to providing a fair and constructive peer-review process. Do not hesitate to contact us if there are specific requests from the reviewers that you believe are technically impossible or unlikely to yield a meaningful outcome.

We hope to receive your revised manuscript within 6-8 weeks. I would be grateful if you could contact us as soon as possible if you foresee difficulties with meeting this target resubmission date.

- Include a "Response to the editors and reviewers" document detailing, point-by-point, how you addressed each editor and referee comment. If no action was taken to address a point, you must provide a compelling argument. When formatting this document, please respond to each reviewer comment individually, including the full text of the reviewer comment verbatim followed by your response to the individual point. This response will be used by the editors to evaluate your revision and sent back to the reviewers along with the revised manuscript.
- Highlight all changes made to your manuscript or provide us with a version that tracks changes.

Link Redacted

We look forward to seeing the revised manuscript and thank you for the opportunity to review your work. Please do not hesitate to contact me if you have any questions or would like to discuss these revisions further.

Sincerely,

[REDACTED]

Reviewer expertise:

Reviewer #1: Australian archaeology, radiocarbon dating

Reviewer #2: Radiocarbon dating, human occupation of high elevation environments, Paleolithic archaeology

REVIEWER COMMENTS:

Reviewer #1:

Remarks to the Author:

Key results:

- The finding of archaeological materials from the Australian high country, c. 1000 m above sea level, dating to c. 20,000 cal BP. This is about twice the age of the previously oldest known archaeological evidence of people at such an elevation in SE Australia.
- This is an important finding that demonstrates through evidence that people occupied the high country that far back in time. Note that most archaeologists thought this was probably the case, but no evidence has existed until now.

Validity:

- There are major flaws, but none are unfixable. The starting sentence in the ms captures multiple problems: "Australia's eastern highlands have traditionally been viewed as a cold-climate barrier to early migrations". (A) What do the authors mean by "eastern highlands". This geographical area usually refers to the Great Dividing Range, a north-south mountain range along Australia's eastern seaboard that spans from the tropics to the SE corner of mainland Australia. It is not clear however what the authors mean by "eastern highlands", and they seem to change the goalposts through the paper, so that it's not clear if they mean the Blue Mountains, Australian Alps, alpine and/or sub-alpine zones generally, high elevation areas including below the sub-alpine zone, etc. (B) Contra- this paper's statement on this, this mountain range (eastern highlands meaning Great Dividing Range) has not been "traditionally viewed as a cold-climate barrier to early migrations". Much of the Great Dividing Range is not even cold-climate country. If they mean the Australian Alps, or alpine and sub-alpine elevations, then they need to say so. (C) More is needed as to the position of the reported site relative to alpine and sub-alpine conditions for each of the time-frames in question for the article (because the elevation of alpine and sub-alpine conditions changed through the course of deep-time). This needs to be carefully referenced throughout, and evidence-based. (D) They refer to "early migrations", but the term "migration" is not well applied here. Migration is only one kind of mobility. Do they mean movements for colonisation and/or settlement away from prior homelands, or other kinds also (seasonal movements, trade, travels etc.). (E) The site in question is about 20,000 years old; how does this fit "early migrations" in a broader region that has already been colonised for over 50,000 years? 20,000 cal BP does not seem "early" in this context ... 30,000 years have already gone by.
- Given the above, it would be much better to simply frame this paper around "High country occupation of SE Australia during the peak of the last ice age" or something to that effect, without creating a false straw person saying that the mountain chain was a cold-climate barrier to early migrations. Rather, emphasise that elevations of around 1000 m above sea level were occupied then (any given location seasonally? Permanently?).
- Please note that the Blue Mountains are not in the Australian Alps, nor in today's alpine or sub-alpine zones. At that latitude, c. 1000 m above sea level is usually considered below the sub-alpine zone. I am unsure as to the exact positioning of this site relative to the tree-line and permanent glaciation around 20,000 cal BP; the article should make this explicit, always through referenced evidence. The issue is that the details are often masked in the article by referring to broad concepts such as "Pleistocene", "Holocene", "LGM", "high country" etc, but in reality in most cases when these concepts are used they mask the particular point in time, or elevation, or association with a particular palaeoclimatic condition etc that is at stake. This flaw is fixable by referring instead to the details (e.g. exact calibrated age, or exact elevation, or exact climatic condition for a particular point in time and specific elevation). This problem of masking the issue or masking the association of issues (e.g. occupation with particular climatic conditions) is not unique to this paper; the use of broad categories rather than specific details gives a semblance of unification (same time, or same place, etc.) when in reality multiple not directly relatable things may be happening, and this is why in this article the details matter.
- All of the above flaws are fixable.
- More concerning are the coarse-grained excavation methods used (in particular the use of thick, 5cm spits and the failure to sub-divide some layers into much thinner excavation units to enable finer chronostratigraphic patterns to emerge, and taphonomic issues to be investigated at a finer scale), which are not fixable. It is imperative that the excavation units are superimposed over the section drawings so that the reader can see the exact correspondence of excavation units against layers (and carbon dates). Nevertheless the presence of people at the site c. 20,000 cal BP is convincing with the evidence at hand.

Originality and significance:

- The major claim of having discovered c. 20,000 year-old occupation at c. 1000 m above sea level in SE Australia is a major finding because we now have actual evidence of people that high up in the high country during the peak of the last ice age. However, the authors don't need to set up a straw person and say that this is contrary to recent or current thinking, as it is not, but rather this finding sets a new evidential foundation.
- I do think that these results are of immediate interest to many people in my own discipline, as well as to people from several other disciplines such as palaeoecology, as well as to the general public.

Data & methodology:

- This paper uses a standard archaeological methodology (albeit the excavation methods used are rather coarse-grained, using 5 cm-thick excavation units). The data presented is sufficient to reveal the presence of people at this site at the times in the past listed (back to c. 20,000 cal BP). Please see my in-text comments. Something that is not sufficiently detailed and transparent to enable reproducing the results is the plotting of the excavation units onto the section drawings. This would enable the reader to see exactly what the correspondences are, where the excavators may have cut across strata, where the stratigraphic interfaces (zones of mixing) are relative to the excavation units (i.e. how the deposit was sampled, and the ages of those samples). Something else that is currently missing but required are details of the depth of stratigraphic interfaces between layers. This is crucial as interfaces are zones of mixing, so understanding how excavation units divided/sampled layers and their interfaces is critical to the reliability of chronostratigraphic results and characterisations.

Preregistration: n/a

Appropriate use of statistics and treatment of uncertainties: Something that needs fixing is how they report/discuss ages/radiocarbon ages. It is fundamental that throughout all parts of the manuscript, consistency is used when reporting uncalibrated radiocarbon ages, individual calibrated (unmodelled) ages, and Bayesian-modelled ages. Don't round-off ages as they are statistical statements. It's important that the reader knows the statistical limits of how to interpret such ages, so sticking by the actual Bayesian modelled age ranges (uncertainty ranges) is important. Currently sometimes ages seem to be variably presented as rounded-off approximates, "years ago", "cal BP", median, uncertainty age ranges etc inconsistently presented (and without the details to know what's what). This needs to be fixed, and is fixable. I suspect once fixed it won't change the main story of the article, but these details need to be seriously sharpened to make the data presentable.

Custom code: n/a

Conclusions: Do you find that the conclusions and data interpretation are robust, valid and reliable? Yes, in the sense that they present important new data as evidence for the occupation of the high country up to about 1000 m above sea level in SE Australia during the peak of the last ice age.

Suggested improvements: A major improvement required, in addition to those listed above (and those listed in Track Changes and inserted marginal comments on the article ms itself) would be to have much sharper, more detailed information about palaeoenvironmental conditions in the immediate area of the reported site, for each of the times when people were present at the site, especially for the specific period of first occupation of the site. This needs to be evidence-based and fully referenced.

References: Missing references include Joanna Fresløv and Russell Mullett (2022) "Below the Sky, Above the Clouds: The Archaeology of the Australian High Country", in *The Oxford Handbook of the Archaeology of Indigenous Australia and New Guinea* (key high country sites and models for high country occupation are published there, but this article is not referenced at all. It is a key paper that I am surprised has been omitted).

Josephine Flood's (1980) *The Moth Hunters* should also be cited early on as an early author on the archaeology of the high country, even if in passing.

Clarity and context: Is the abstract clear, accessible? Are abstract, introduction and conclusions appropriate? Yes, but see my comments above.

Please indicate any particular part of the manuscript, data, or analyses that you feel is outside the scope of your expertise, or that you were unable to assess fully: I am not able to fully assess the climatic and environmental conditions for the periods of times in question in the article. I suggest an active palaeoclimatologist and palaeoecologist for SE Australia be asked to assess that dimension of the paper. Professor Simon Haberle at the Australian National University would be apt. I am also unable to sufficiently assess the pXRF methodology. Dr Jillian Huntley at Griffith University would be apt.

Reviewer #2:

Remarks to the Author:

This is a well-conceived manuscript that provides original research with a relevant potential to make a significant advance in the field of Palaeolithic Archaeology and our understanding of settlement patterns and human-climate-environment interactions during the Last Glacial Maximum in Australia. The authors present new evidence showing that glacial landscapes were not always a barrier for human occupation during the Late Pleistocene and they show the oldest evidence of high-elevation (above 1000 m) Pleistocene occupation in Australia. Therefore, the relevance of the research to be published is unquestionable.

Research questions have been correctly devised and results are discussed accordingly. The article is well structured in general terms, and the text is well written, Therefore, the manuscript deserves publication in a high-quality scientific journal.

However, to my view, the authors should deal with several issues, some of them relevant, before considering the manuscript for publication. Most of these issues are related to the presentation and discussion of the data, and in some cases to their quality and quantity. For the sake of clarity, I will follow the order of the Methods section to discuss my criticism:

Charcoal sample collection

In lines 320-325 the authors write "As timber was not present in this landscape during the LGM, charcoal in layers 9-15 was most likely introduced to the site through anthropogenic fires, however, charcoal fragments could also have been blown in from distant cultural burning/wildfires. Hearth charcoal was differentiated from potential cultural burning/wildfire charcoal by its consolidation in bounded concentrations and association with burnt sediment and/or ash and/or heat damaged artefacts."

Two remarks on this:

(1) why the authors have not conducted pollen, anthracological or other paleoenvironmental analyses to provide solid evidence on the landscape and vegetation reconstruction of the area throughout the sequence of human occupation registered at the site? At some points in the manuscript they cite regional pollen sequences (line 140), but it is unclear whether such data are sound enough to provide a detailed reconstruction of the vegetation history of the study area. Direct palaeoecological evidence collected from the site's stratigraphic sequence would have provided more valuable evidence to discuss human-environment interactions throughout the sequence of human occupation at the site.

(2) They mention the identification of hearths (fire structures) in different stratigraphic layers. However, not a single photo of these hearths is provided in the manuscript or the supplementary material. Not even in the case of sample 74930, which is the one dating the purported oldest human occupation recorded at the site at ~20000 years ago in level 15. Given the very scarce lithic component of this layer (as it will be discussed below), the authors should provide solid evidence demonstrating that this sample really comes from a hearth, and hence is clearly related to human activity (and not to charcoal fragments "blown in from distant cultural burning/wildfires" as the authors themselves acknowledge that could be the case for some samples).

Radiocarbon dating and modelling

Besides the issue related to the discussion of sample collection from hearths discussed above, the dated samples are probably too few to assure a robust chronology for the sequence of human occupation. This is especially the case for the oldest occupation recorded in level 14-15, for which only one date is available. The authors should consider conducting more radiocarbon assays to gain statistical significance.

In Supplementary Fig. 1 and Fig. 4 the sample S-ANU 74930 (shown here as the unmodeled calibrated date of 20296) is located in layer 14. However, in the Supplementary Table 3, Supplementary figure 3 and Fig. 3 this sample is related to layer 15. Where is the mistake? In general terms, the authors should make a greater effort to present their data more clearly. If the date belongs to layer 15, then the artifacts from layer 14 are younger than 20296 years ago.

Chemical results of each dated sample should be provided.

Both the code and the raw data of the chronological models should be provided.

In Supplementary Fig. 2 it is shown the location of two OSL samples in two stratigraphic profiles. However, no mention of such samples and their potential results is made in the manuscript or the supplementary material. Why?

Finally, the authors should make clear if sample 74932, collected in layer 18 and dated to 53426 cal years BP, is interpreted as evidence of human presence or not. It seems it is not (lines 115-117), but then it is even more necessary to demonstrate that the other dated charcoals are actually related to human activity, given that this one was introduced to the site by natural processes.

Artefact analysis

The lithic artefacts should be clearly counted per stratigraphic layer, including a basic assignment of each of them to a technological category as described by the authors. With the provided information, one can only have partial views of the distribution of the lithics throughout the sequence according to fig. 4 and some parts of the manuscript. This is not enough, as quantitative and concise data should be provided in a clear manner. This is especially relevant to assess if the proposed interpretations concerning the oldest human occupation recorded at the site are robust. How many lithic artefacts are related to radiocarbon sample S-ANU 74930? Are they found in layer 14 or layer 15? and what is their exact spatial relation to such sample? (see comments above, including the necessity of providing graphical evidence of the hearth where this sample is claimed to come from). A more concise and clear presentation of all these data is needed, especially considering the scarce nature of the lithic assemblage (four artefacts putting together layers 14 and 15 as described in lines 158 and 159?) and the existence of only one radiocarbon date related to this occupation.

Furthermore, the presentation of the lithics in Supplementary Fig. 5 is odd. The artefacts are assembled according to their raw material, but to my view they should be first assembled according to their stratigraphic position (not mixing Holocene and Pleistocene artefacts).

Rock art analysis

The manuscript provides a description of methods for studying rock art at Dargan shelter. However, the obtained results or the significance of this art is barely discussed in the text. What is the proposed relation between the art and the human presence at this site? Why haven't they tried to radiocarbon date the black paintings (as they were made with charcoal)?

Finally, another relevant issue, which is pertinent throughout the text, is the lack of a proper definition of the Last Glacial Maximum and its chronological limits. Given the ongoing discussions on the definition, extent and chronology of the LGM and their differences around the world, the authors should clarify what do the mean by 'LGM' and its significance in the Australian context (see e.g. Cadd, H., Williams, A.N., Saktura, W.M. et al. Last Glacial Maximum cooling induced positive moisture balance and maintained stable human populations in Australia. *Commun Earth Environ* 5, 52 (2024). <https://doi.org/10.1038/s43247-024-01204-1>). In this sense, in lines 180 and 181 the authors write 'Four of these 'unidentified silicious' flakes were found in layers 11 and 12, which date to 16-17,000 years ago, suggesting a southern entry to the region during the height of the LGM.' However, the most common definition of the LGM situates its peak between 23000 and 19000 years ago, making the authors' statement in need of an explanation or correction.

All in all, although there is no doubt that the research is original and presents novel and significant results of interest to a wide variety of researchers, an incomplete and not very clear presentation of the data, together with the necessity of providing more robust evidence (i.e. more chronometric dates and paleoenvironmental data collected at the site) are major flaws. To my view, only if these issues are corrected the manuscript deserves to be considered for publication in *Nature Human Behaviour*.

Other minor issues are the following:

Lines 41-43: The mentioned geographic location should be linked to a map, especially considering readers not familiar with the Australian geography.

Fig.2 should be enlarged.

Reference 2, cited as "Río J. A. et al." should be cited as "Arangocillo-del Río J. et al"

Supplementary Table 2: Is "Cal BC" (and not Cal BP) correct in the caption?

Supplementary Fig. 6: In the caption, "[...]the Holocene-Pleistocene tradition" should be read as "[...]the Holocene-Pleistocene

transition”.

Version 1:

Decision Letter:

14th November 2024

Dear Dr Way,

Thank you once again for your manuscript, entitled "The earliest evidence of high-elevation ice age occupation in Australia," and for your patience during the peer review process.

Your manuscript has now been evaluated by the 2 reviewers from the previous round, whose comments are included at the end of this letter. We are very interested in the possibility of publishing your study in Nature Human Behaviour, but would like to consider your response to the Reviewer's remaining concerns in the form of a revised manuscript before we make a decision on publication. In particular we ask that you address Reviewer 1's outstanding concerns in full.

In sum, we invite you to revise your manuscript taking into account all reviewer and editor comments. We are committed to providing a fair and constructive peer-review process. Do not hesitate to contact us if there are specific requests from the reviewers that you believe are technically impossible or unlikely to yield a meaningful outcome.

We hope to receive your revised manuscript within 2-3 weeks. I would be grateful if you could contact us as soon as possible if you foresee difficulties with meeting this target resubmission date.

- Include a "Response to the editors and reviewers" document detailing, point-by-point, how you addressed each editor and referee comment. If no action was taken to address a point, you must provide a compelling argument. When formatting this document, please respond to each reviewer comment individually, including the full text of the reviewer comment verbatim followed by your response to the individual point. This response will be used by the editors to evaluate your revision and sent back to the reviewers along with the revised manuscript.
- Highlight all changes made to your manuscript or provide us with a version that tracks changes.

Link Redacted

We look forward to seeing the revised manuscript and thank you for the opportunity to review your work. Please do not hesitate to contact me if you have any questions or would like to discuss these revisions further.

Sincerely,

[REDACTED]

REVIEWER COMMENTS:

Reviewer #1 (Remarks to the Author):

I have now reviewed the revised ms (and the response to the original reviewers' comments). I think the authors have done an excellent job responding to most of the reviewers' comments. Please note that my point about the use of terms for major categories isn't entirely addressed in-text (e.g. referring to things to the "Pleistocene" isn't really useful, as that's such a broad span of time; much better to specify the particular time-frame at stake, e.g. refer to an envelope of time). I think there are still 2 other major (but easily addressed) and 3 minor (some stylistic) comments to address:

Major points:

Please explicitly tell the reader what the thickness of the stratigraphic interfaces are for each layer. For example, "the transition from Layer 1 to Layer 2 is gradual, taking place over a depth of 2 cm". This is important because it tells the reader what the visible mixed zone between those layers are (interfaces, and their thicknesses, should always be commented on when reporting a stratigraphic sequence). Please note that this is not about the vertical distribution of artefacts (excellent to see these plotted on the section drawings; while it is standard to record artefact locations in 3 dimensional space in Australian archaeology, too rarely do researchers publish their locations on section drawings), but of the sediments themselves. For example, what is the thickness of the interface between Layers 9 and 10 (and why have some layers been distinguished by black lines on Figure 4, but not others? Again on Figure 4, why is there a subscript 4 after the median age of 5525 ? (it would be much easier to read the ages if from 10,000 onwards the thousands were separated by commas throughout the paper including all the figures and tables).

Please plot on the section drawings the excavation units. This is not about relating the 3D-plotted artefacts onto the section drawings, but additional to this, as it's very important for the reader to see how the excavation levels map onto (or don't) the stratigraphy. There will be times when an excavation unit cuts across a stratigraphic interface, for example, so while it's common to excavate stratigraphic interfaces separately (as their own excavation unit, to make sure there's no contamination of the materials included in a particular excavated stratigraphic layer), this doesn't always happen (and often stratigraphic interfaces can only be seen clearly AFTER an excavation is finished). The reader needs to be able to see how (1) excavation units, (2) stratigraphic layers, (3) in situ finds (the artefact dots presented in this paper), and (4) the carbon dates all are in relation to each other, on the section drawing. This is particularly important here given the paucity of stone artefacts in the lower levels where they occur. (I assume excavation units exist given that they refer to them e.g. in the caption of Supplementary Figure 4). If, on the other hand, the excavation did not subdivide layers into thinner excavation units, then please explicitly tell the reader so, and that stratigraphic interfaces were combined with either the overlying or underlying stratigraphic layer proper (in which case the chronostratigraphy cannot be further assessed by looking at the vertical pattern(s) and contexts of the excavated layers, sediments and their finds).

Minor points:

I suggest using "cal BP" or "cal. BP" throughout, not "cal. yrs BP". The "yrs" is superfluous, and "cal BP" or "cal. BP" is more consistent with standard archaeological conventions.

Please standardise to "Dargan Shelter" throughout; at least one other time I read "Dargan rock shelter".

Note that Late Pleistocene, Early Holocene, Mid-Holocene and Late Holocene should all be capitalised, as they have all been ratified by the International Commission on Stratigraphy (see <https://stratigraphy.org/ICSchart/ChronostratChart2023-09.pdf>).

Once these issues are addressed, I recommend the paper be published; it presents exciting archaeological findings, and is of both national and international, academic and popular interest.

Reviewer #2 (Remarks to the Author):

The authors have sufficiently addressed most of my comments and requests, and together with the changes implemented as a result of the other reviewer's comments, the manuscript has been significantly improved. I still think that the data supporting the human presence around 20,000 cal BP could be stronger, as only 2 lithic pieces belong to level 14 (plus another 2 in level 15), the hearth recorded in such layer is not so evident according to the provided photo, and only 1 radiocarbon date is available for that layer. Furthermore, the 4 mentioned lithic pieces are not identified among those photographed in Supplementary fig. 8 and I think that, at least those recorded in level 14, should be shown either by photos or drawings.

Anyhow, even if these points cannot be addressed (as they were not in the last submission), I think the research deserves publication in a high quality scientific journal.

Version 2:

Decision Letter:

Our ref: NATHUMBEHAV-24072795B

4th December 2024

Dear Dr. Way,

Thank you for submitting your revised manuscript "The earliest evidence of high-elevation ice age occupation in Australia" (NATHUMBEHAV-24072795B). We have now reviewed the revised manuscript editorially. Thank you for your patience during this process.

I am delighted to tell you that we will be happy in principle to publish it in Nature Human Behaviour, pending minor revisions to comply with our editorial and formatting guidelines.

We are now performing detailed checks on your paper and will send you a checklist detailing our editorial and formatting requirements within two weeks. Please do not upload the final materials and make any revisions until you receive this additional information from us.

Sincerely,

[REDACTED]

Version 3:

Decision Letter:

Dear Dr Way,

We are pleased to inform you that your Article "The earliest evidence of high-elevation ice age occupation in Australia", has now been accepted for publication in Nature Human Behaviour.

With best regards,

[REDACTED]

P.S. Click on the following link if you would like to recommend Nature Human Behaviour to your librarian
<http://www.nature.com/subscriptions/recommend.html#forms>

** Visit the Springer Nature Editorial and Publishing website at http://editorial-jobs.springernature.com?utm_source=ejp_NHumB_email&utm_medium=ejp_NHumB_email&utm_campaign=ejp_NHumB for more information about our career opportunities. If you have any questions please click [here](mailto:editorial.publishing.jobs@springernature.com).

Reviewer #1:

Validity

- There are major flaws, but none are unfixable. The starting sentence in the ms captures multiple problems: “Australia’s eastern highlands have traditionally been viewed as a cold-climate barrier to early migrations”. (A) What do the authors mean by “eastern highlands”. This geographical area usually refers to the Great Dividing Range, a north-south mountain range along Australia’s eastern seaboard that spans from the tropics to the SE corner of mainland Australia. It is not clear however what the authors mean by “eastern highlands”, and they seem to change the goalposts through the paper, so that it’s not clear if they mean the Blue Mountains, Australian Alps, alpine and/or sub-alpine zones generally, high elevation areas including below the sub-alpine zone, etc.

RESPONSE: Australia’s Eastern Highlands are also known as the Great Dividing Range and include the Blue Mountains and the Australian Alps. We have addressed the ambiguity noted by reviewer 1 by adding a new figure (Supplementary Fig. 1) which shows the location of the Pleistocene sites mentioned in the text in relation to the Eastern Highlands; adding a reference which establishes the extent of the Eastern Highlands to Supplementary Information (ref 1); and by capitalising Eastern Highlands to signal that we are referring to the named region, rather than utilising a descriptive term.

- (B) Contra- this paper’s statement on this, this mountain range (eastern highlands meaning Great Dividing Range) has not been “traditionally viewed as a cold-climate barrier to early migrations”. Much of the Great Dividing Range is not even cold-climate country. If they mean the Australian Alps, or alpine and sub-alpine elevations, then they need to say so.

RESPONSE: We are referring to the elevated regions of the Eastern Highlands/Great Dividing Range during the Pleistocene, which were significantly cooler then. The paucity of archaeological data from this region and time underpins the cold climate barrier model, which is currently under debate in Australian archaeology, with some archaeologists conceptualising the Blue Mountains as a barrier to early mobility (Williams et al 2014) and the Great Dividing Range as an LGM barrier (Barry et al 2021). Other archaeologists have posited that the eastern highlands (uplands) were entirely unoccupied in the Pleistocene (Hiscock and Sterelny 2023). These references have been added to the manuscript p. 2 (refs 17-19).

- (C) More is needed as to the position of the reported site relative to alpine and sub-alpine conditions for each of the time-frames in question for the article (because the elevation of alpine and sub-alpine conditions changed through the course of deep-time). This needs to be carefully referenced throughout, and evidence-based.

RESPONSE: We have added two additional lines of supporting evidence. First, we have directly counted pollen grains on the LGM sediments (see new Supplementary Table 4) and added the following text to the manuscript, p.4, “Pollen analyses, although with low counts, indicate no arboreal pollen at the site and regional pollen records indicate the treeline was ~400 m below the site”. ‘Pollen Analysis’ has been added to Methods. Secondly, we have added two additional references supporting regional vegetation interpretations to the manuscript (refs 23,24). Unfortunately, there

are insufficient regional sites to specifically place the altitude of the treeline for every interval. There is only a brief interval where this is not constrained, before arboreal pollen is significant at Dargan Shelter. However, this does not affect our primary conclusion of the presence of occupation during the LGM.

- (D) They refer to “early migrations”, but the term “migration” is not well applied here. Migration is only one kind of mobility. Do they mean movements for colonisation and/or settlement away from prior homelands, or other kinds also (seasonal movements, trade, travels etc.).

RESPONSE: We have changed the term ‘migration’ to ‘mobility’ or ‘movement’ throughout.

- (E) The site in question is about 20,000 years old; how does this fit “early migrations” in a broader region that has already been colonised for over 50,000 years? 20,000 cal BP does not seem “early” in this context ... 30,000 years have already gone by. Given the above, it would be much better to simply frame this paper around “High country occupation of SE Australia during the peak of the last ice age” or something to that effect, without creating a false straw person saying that the mountain chain was a cold-climate barrier to early migrations. Rather, emphasise that elevations of around 1000 m above sea level were occupied then (any given location seasonally? Permanently?).

RESPONSE: We have replaced the term ‘early migrations’ with ‘movement during the LGM’ and adopted the suggestion to frame the paper around occupation during the peak of the last ice age, see tracked changes in Discussion and Conclusions.

- Please note that the Blue Mountains are not in the Australian Alps, nor in today’s alpine or sub-alpine zones. At that latitude, c. 1000 m above sea level is usually considered below the sub-alpine zone. I am unsure as to the exact positioning of this site relative to the tree-line and permanent glaciation around 20,000 cal BP; the article should make this explicit, always through referenced evidence. The issue is that the details are often masked in the article by referring to broad concepts such as “Pleistocene”, “Holocene”, “LGM”, “high country” etc, but in reality in most cases when these concepts are used they mask the particular point in time, or elevation, or association with a particular palaeoclimatic condition etc that is at stake. This flaw is fixable by referring instead to the details (e.g. exact calibrated age, or exact elevation, or exact climatic condition for a particular point in time and specific elevation). This problem of masking the issue or masking the association of issues (e.g. occupation with particular climatic conditions) is not unique to this paper; the use of broad categories rather than specific details gives a semblance of unification (same time, or same place, etc.) when in reality multiple not directly relatable things may be happening, and this is why in this article the details matter. All of the above flaws are fixable.

RESPONSE: We have added that the site is below the modern subalpine zone in the manuscript (see site description p. 3) and added the location of the Blue Mountains and Australian Alps relative to the sites mentioned in the text (new Supplementary Fig. 1).

We have removed references to undefined or broad concepts. Specifically, we have removed “ice age” (except for the title, where it remains for purposes of accessibility),

“uplands”, “highlands”, “glacial period” and terms qualifying the LGM where these are ambiguous. With regard to “I am unsure as to the exact positioning of this site relative to the tree-line and permanent glaciation around 20,000 cal BP”, these elevations are given in figure 1 and Supplementary Fig. 1, which locates Mount Kosciuszko where the glaciation occurred.

- More concerning are the coarse-grained excavation methods used (in particular the use of thick, 5cm spits and the failure to sub-divide some layers into much thinner excavation units to enable finer chronostratigraphic patterns to emerge, and taphonomic issues to be investigated at a finer scale), which are not fixable. It is imperative that the excavation units are superimposed over the section drawings so that the reader can see the exact correspondence of excavation units against layers (and carbon dates). Nevertheless the presence of people at the site c. 20,000 cal BP is convincing with the evidence at hand.

RESPONSE: The reviewer expands on this comment under ‘Data and methodology’, below. A detailed response is provided there.

Originality and significance

- The major claim of having discovered c. 20,000 year-old occupation at c. 1000 m above sea level in SE Australia is a major finding because we now have actual evidence of people that high up in the high country during the peak of the last ice age. However, the authors don’t need to set up a straw person and say that this is contrary to recent or current thinking, as it is not, but rather this finding sets a new evidential foundation.

RESPONSE: Text changed in Discussion and Conclusions (see track changes) to address this comment, including the deletion of “It is now clear that the Eastern Australian Highlands did not pose a barrier to early mobility”.

- I do think that these results are of immediate interest to many people in my own discipline, as well as to people from several other disciplines such as palaeoecology, as well as to the general public.

Data & methodology

- This paper uses a standard archaeological methodology (albeit the excavation methods used are rather coarse-grained, using 5 cm-thick excavation units). The data presented is sufficient to reveal the presence of people at this site at the times in the past listed (back to c. 20,000 cal BP). Please see my in-text comments. Something that is not sufficiently detailed and transparent to enable reproducing the results is the plotting of the excavation units onto the section drawings. This would enable the reader to see exactly what the correspondences are, where the excavators may have cut across strata, where the stratigraphic interfaces (zones of mixing) are relative to the excavation units (i.e. how the deposit was sampled, and the ages of those samples). Something else that is currently missing but required are details of the depth of stratigraphic interfaces between layers. This is crucial as interfaces are zones of mixing, so understanding how excavation units

divided/sampled layers and their interfaces is critical to the reliability of chronostratigraphic results and characterisations.

RESPONSE: Here reviewer 1 requests additional detail and data display to more clearly show the relationship between samples and stratigraphic layers.

As our methods involved a multi-square excavation which employed 3D piece plotting, we have addressed this comment by following data display protocols established in ^{1,2}.

Specifically, we have:

- 1. updated the methods-Archaeological Excavation section p. 9 to clearly outline that piece plotting rather than bulk excavation was employed, and*
- 2. added 2 new section drawings (Supplementary Figs. 4 and 5) which plot the dated samples and artefacts in relation to the stratigraphic layers in the main trench. This will help the reader to clearly see exactly what the correspondences are between artefacts, stratigraphic layers and dated samples and addresses the request for additional data and improved clarity in the display of that data.*

Appropriate use of statistics and treatment of uncertainties

- Something that needs fixing is how they report/discuss ages/radiocarbon ages. It is fundamental that throughout all parts of the manuscript, consistency is used when reporting uncalibrated radiocarbon ages, individual calibrated (unmodelled) ages, and Bayesian-modelled ages. Don't round-off ages as they are statistical statements. It's important that the reader knows the statistical limits of how to interpret such ages, so sticking by the actual Bayesian modelled age ranges (uncertainty ranges) is important. Currently sometimes ages seem to be variably presented as rounded-off approximates, "years ago", "cal BP", median, uncertainty age ranges etc inconsistently presented (and without the details to know what's what). This needs to be fixed, and is fixable. I suspect once fixed it won't change the main story of the article, but these details need to be seriously sharpened to make the data presentable.

RESPONSE: We have standardised the language and removed inconsistent terminology of "years ago", "ka" etc, however we do keep some general language in the abstract to make it more accessible.

Ages are rounded off as it is normal practise within geochronology to round ages to the precision of the dating (e.g. Oxford radiocarbon date lists). Here we have rounded age to the nearest decade because it is not possible to date within 1 year 20,000 years ago. The Bayesian statistical probabilities are decadal at best. To address this, we added the age ranges (Unmodelled (cal.BP 95.4%)) as presented in the main manuscript and Supplementary Tables 2 and 3. Supplementary Table 3 now lists the radiocarbon ages, the unmodelled dates and the FC14 values and error margins for our own dates as requested by the reviewer. To preserve the precision of the radiocarbon measurements for future recalibration, we have included the original F14C measurements to four significant figures.

Conclusions

- Do you find that the conclusions and data interpretation are robust, valid and reliable? Yes, in the sense that they present important new data as evidence for the

occupation of the high country up to about 1000 m above sea level in SE Australia during the peak of the last ice age.

Suggested improvements

- A major improvement required, in addition to those listed above (and those listed in Track Changes and inserted marginal comments on the article ms itself) would be to have much sharper, more detailed information about palaeoenvironmental conditions in the immediate area of the reported site, for each of the times when people were present at the site, especially for the specific period of first occupation of the site. This needs to be evidence-based and fully referenced.

RESPONSE: We have added a summary of the currently available pollen information to Supplementary Text. We note serious issues with chronology and dating of palaeoenvironmental archives from the region, particularly for the Pleistocene. To supplement this, we have provided pollen counts (Supplementary Table 4) for four samples from the archaeological sequence at Dargan. Pollen concentrations are extremely low. An absence of arboreal vegetation may be indicated in the Pleistocene levels (grass pollen only type counted >1), however the samples may not be representative of the full local vegetation. The particle size analysis indicates the primary sedimentary input is sand-sized particles spalling from the rock shelter and the catchment of any environmental signal is likely limited. Additionally, these deposit types are not well-suited to the preservation of pollen. Through our data and the additional Supplementary text & Supplementary Table 4 we have provided the best-available information on the regional paleoenvironmental conditions.

References

- Missing references include Joanna Fresløv and Russell Mullett (2022) “Below the Sky, Above the Clouds: The Archaeology of the Australian High Country”, in The Oxford Handbook of the Archaeology of Indigenous Australia and New Guinea (key high country sites and models for high country occupation are published there, but this article is not referenced at all. It is a key paper that I am surprised has been omitted).

RESPONSE: This reference has been added (manuscript ref. 11).

- Josephine Flood’s (1980) The Moth Hunters should also be cited early on as an early author on the archaeology of the high country, even if in passing.

RESPONSE: This reference has been added (manuscript ref. 12).

Reviewer #2:

- This is a well-conceived manuscript that provides original research with a relevant potential to make a significant advance in the field of Palaeolithic Archaeology and our understanding of settlement patterns and human-climate-environment interactions during the Last Glacial Maximum in Australia. The authors present new evidence showing that glacial landscapes were not always a barrier for human occupation during the Late Pleistocene and they show the oldest evidence of high-elevation (above 1000 m) Pleistocene occupation in Australia. Therefore, the relevance of the research to be published is unquestionable.
- Research questions have been correctly devised and results are discussed accordingly. The article is well structured in general terms, and the text is well written, Therefore, the manuscript deserves publication in a high-quality scientific journal.
- However, to my view, the authors should deal with several issues, some of them relevant, before considering the manuscript for publication. Most of these issues are related to the presentation and discussion of the data, and in some cases to their quality and quantity. For the sake of clarity, I will follow the order of the Methods section to discuss my criticism:

Charcoal sample collection

- In lines 320-325 the authors write “As timber was not present in this landscape during the LGM, charcoal in layers 9-15 was most likely introduced to the site through anthropogenic fires, however, charcoal fragments could also have been blown in from distant cultural burning/wildfires. Hearth charcoal was differentiated from potential cultural burning/wildfire charcoal by its consolidation in bounded concentrations and association with burnt sediment and/or ash and/or heat damaged artefacts.”

Two remarks on this:

- (1) why the authors have not conducted pollen, anthracological or other palaeoenvironmental analyses to provide solid evidence on the landscape and vegetation reconstruction of the area throughout the sequence of human occupation registered at the site? At some points in the manuscript they cite regional pollen sequences (line 140), but it is unclear whether such data are sound enough to provide a detailed reconstruction of the vegetation history of the study area. Direct palaeoecological evidence collected from the site’s stratigraphic sequence would have provided more valuable evidence to discuss human-environment interactions throughout the sequence of human occupation at the site.

RESPONSE: We have added a more comprehensive review of regional palaeoenvironmental records to Supplementary Text-Pollen records to provide additional context for human-environment interaction. We have added pollen evidence from the site in Supplementary Table 4, noting that this deposit is not well-suited to the preservation of plant microfossils. See also the detailed response to reviewer 1 above.

- (2) They mention the identification of hearths (fire structures) in different stratigraphic layers. However, not a single photo of these hearths is provided in the manuscript or the supplementary material. Not even in the case of sample 74930, which is the one dating the purported oldest human occupation recorded at the site at ~20000 years ago in level 15. Given the very scarce lithic component of this layer (as it will be discussed below), the authors should provide solid evidence demonstrating that this sample really comes from a hearth, and hence is clearly related to human activity (and not to charcoal fragments “blown in from distant cultural burning/wildfires” as the authors themselves acknowledge that could be the case for some samples).

RESPONSE: We have added a photo of the lowest hearth and dated charcoal from this hearth (Supplementary Fig. 14) and updated Supplementary Table 3 to indicate more clearly the origin of each charcoal sample.

Radiocarbon dating and modelling

- Besides the issue related to the discussion of sample collection from hearths discussed above, the dated samples are probably too few to assure a robust chronology for the sequence of human occupation. This is especially the case for the oldest occupation recorded in level 14-15, for which only one date is available. The authors should consider conduct more radiocarbon assays to gain statistical significance.

RESPONSE: Although reviewer #2 suggests that too few samples were dated, we disagree as the 24 dated samples provide the best dated record in high altitude Australia. For comparative purposes, the radiocarbon chronology of Australia’s oldest site was built from 22 charcoal samples² and the oldest site in the Sydney basin was dated from 25 OSL samples as no charcoal or other suitable material was identified³.

In addition, we consider the chronology to be extremely robust for Dargan Rockshelter. This is reflected in the two Bayesian models in OxCal we have used to test the dating sequence. Both models demonstrate excellent agreement. Of most relevance for understanding agreements between age and stratigraphic sequence is the General T-type model (Model A in our discussion). As we have added to the text, this model, Model A (Fig. 3), shows that all but sample S-ANU74937 (with a 6% value) have a posterior outlier probability values of 5% or less. This has two important implications. Firstly, it illustrates that none that would suggest any of the dates demonstrate an ‘old wood’ effect, and/or are too young or too old for their respective locations within the within the chronological/stratigraphic sequence. Secondly, it indicates that there is not a single age inversion in the stratigraphic sequence. This illustrates the integrity of the archaeological stratigraphy and the quality of sample selection.

To address this concern, we have added information on the additional samples that were submitted for dating, but which proved unsuitable to Supplementary Table 3. In total, 32 potential samples were submitted to the Australian National University and UNSW radiocarbon laboratories for dating. Of these, eight samples were considered unsuitable for dating as they consisted of concreted ash with pebbles and small black fragments that were not charcoal. One of these samples was from Holocene Layer 8 and seven samples from the basal Pleistocene Layers 15-18. Unfortunately, this left just one suitable sample from Layer 14 for dating and no further material is available from the basal Pleistocene layers.

- In Supplementary Fig. 1 and Fig. 4 the sample S-ANU 74930 (shown here as the unmodeled calibrated date of 20296) is located in layer 14. However, in the Supplementary Table 3, Supplementary figure 3 and Fig. 3 this sample is related to layer 15. Where is the mistake? In general terms, the authors should make a greater effort to present their data more clearly. If the date belongs to layer 15, then the artifacts from layer 14 are younger than 20296 years ago.

RESPONSE: This has been corrected. The date S-ANU74930 is located in layer 14. Supplementary Table 3, Supplementary figure 3 (now Supplementary Fig. 6) and Fig. 3 have been updated.

- Chemical results of each dated sample should be provided.

RESPONSE: We are not quite sure what the reviewer is requesting here. If the reviewer means the isotopic data, modern accelerator laboratories no longer report the $\delta^{13}\text{C}$ data because of the way the data reduction is performed. If the reviewer is referring to the fraction of modern carbon ($F_{14}\text{C}$) and its error margins we have updated our Supplementary Tables 2 and 3. Table S2 now possesses the modelled ages for Model A (General T-type model) and Model B (Charcoal Plus model), and Table S3 provides information on the uncalibrated, calibrated (unmodelled dates) and $F_{14}\text{C}$ and error margins listed by S-ANU and UNSW- sample number, pit and depth. It also reports on whether the charcoal was recovered from a hearth or just in situ within the deposits.

- Both the code and the raw data of the chronological models should be provided.

RESPONSE: We have added code for the the OxCAL codes for the General T-type and Charcoal Plus models used to model the Dargan rock shelter dates to the Supplementary Information in the new Code - Age-models section.

- In Supplementary Fig. 2 it is shown the location of two OSL samples in two stratigraphic profiles. However, no mention of such samples and their potential results is made in the manuscript or the supplementary material. Why?

RESPONSE: These two samples were collected in case no carbon samples were found in the sterile layers. As a datable sample was found, these OSL samples were not processed. To avoid confusion, they have been removed from Supplementary Fig. 2.

- Finally, the authors should make clear if sample 74932, collected in layer 18 and dated to 53426 cal years BP, is interpreted as evidence of human presence or not. It seems it is not (lines 115-117), but then it is even more necessary to demonstrate that the other dated charcoals are actually related to human activity, given that this one was introduced to the site by natural processes.

RESPONSE: Sample 74932, collected in layer 18, was not interpreted as evidence of human presence, manuscript p.4. The text has been updated to reflect this. The key marker of human activity was the presence of stone artefacts. Additional information has been provided in the form of hearth fig and dwg to demonstrate that this dated charcoal related to human activity. In addition, a column has been added to Supplementary Table 3 which details whether the dated charcoal samples were from hearths or piece plotted samples.

Artefact analysis

- The lithic artefacts should be clearly counted per stratigraphic layer, including a basic assignment of each of them to a technological category as described by the authors. With the provided information, one can only have partial views of the distribution of the lithics throughout the sequence according to fig. 4 and some parts of the manuscript. This is not enough, as quantitative and concise data should be provided in a clear manner. This is especially relevant to assess if the proposed interpretations concerning the oldest human occupation recorded at the site are robust. How many lithic artefacts are related to radiocarbon sample S-ANU 74930? Are they found in layer 14 or layer 15? and what is their exact spatial relation to such sample? (see comments above, including the necessity of providing graphical evidence of the hearth where this sample is claimed to come from). A more concise and clear presentation of all these data is needed, especially considering the scarce nature of the lithic assemblage (four artefacts putting together layers 14 and 15 as described in lines 158 and 159?) and the existence of only one radiocarbon date related to this occupation.

RESPONSE: A table has been added (Supplementary Table 5) which counts the lithic artefacts in each stratigraphic layer and assigns them to a technological category, and by raw material. In addition, 2 new section drawings (Supplementary Figs. 4 and 5) have been added which show the relationship between the stone artefacts, the stratigraphic layers and the dated samples for the main Trench 2.

- Furthermore, the presentation of the lithics in Supplementary Fig. 5 is odd. The artefacts are assembled according to their raw material, but to my view they should be first assembled according to their stratigraphic position (not mixing Holocene and Pleistocene artefacts).

RESPONSE: Supplementary Fig. 5 (now Supplementary Fig. 8) has been amended to assemble the artefacts firstly into Holocene and Pleistocene groups, and then secondarily into raw material groups.

Rock art analysis

- The manuscript provides a description of methods for studying rock art at Dargan shelter. However, the obtained results or the significance of this art is barely discussed in the text. What is the proposed relation between the art and the human presence at this site? Why haven't they tried to radiocarbon date the black paintings (as they were made with charcoal)?

RESPONSE: the rock art is very minimal and degraded and there is insufficient charcoal present for dating or assigning a relation with the human presence at this site. It is very unlikely to be related to the first phase of occupation but has been included in line with the wishes of the Indigenous community, who felt strongly that the presence of rock art, albeit minimal and degraded, in the shelter should be noted.

The manuscript has been updated (Methods – rock art analysis, p. 16) through the addition of: "Insufficient charcoal is present for dating or assigning a relation

with the human presence at this site, however, it is very unlikely to be related to the first phase of occupation.”

- Finally, another relevant issue, which is pertinent throughout the text, is the lack of a proper definition of the Last Glacial Maximum and its chronological limits. Given the ongoing discussions on the definition, extent and chronology of the LGM and their differences around the world, the authors should clarify what do the mean by ‘LGM’ and its significance in the Australian context (see e.g. Cadd, H., Williams, A.N., Saktura, W.M. et al. Last Glacial Maximum cooling induced positive moisture balance and maintained stable human populations in Australia. *Commun Earth Environ* 5, 52 (2024). <https://doi.org/10.1038/s43247-024-01204-1>). In this sense, in lines 180 and 181 the authors write ‘Four of these ‘unidentified silicious’ flakes were found in layers 11 and 12, which date to 16-17,000 years ago, suggesting a southern entry to the region during the height of the LGM.’ However, the most common definition of the LGM situates its peak between 23000 and 19000 years ago, making the authors’ statement in need of an explanation or correction.

RESPONSE: We have now defined the Last Glacial Maximum in the main manuscript text (ref. 15).

- All in all, although there is no doubt that the research is original and present novel and significant results of interest to a wide variety of researchers, an incomplete and not very clear presentation of the data, together with the necessity of providing more robust evidence (i.e. more chronometric dates and palaeoenvironmental data collected at the site) are major flaws. To my view, only if these issues are corrected the manuscript deserves to be considered for publication in *Nature Human Behaviour*.

RESPONSE: This comment aligns with Reviewer #1’s requests for clarity in the presentation of data, additional dating information and additional palaeoenvironmental data, addressed above.

Other minor issues are the following:

- Lines 41-43: The mentioned geographic location should be linked to a map, especially considering readers not familiar with the Australian geography.

RESPONSE: A new figure has been added – Supplementary Figure 1, which shows the location of the sites on a map of south-eastern Australia.

- Fig.2 should be enlarged.

RESPONSE: Figure 2 has been enlarged

- Reference 2, cited as “Río J. A. et al.” should be cited as “Arangocillo-del Río J. et al”

RESPONSE: Citation has been updated.

- Supplementary Table 2: Is “Cal BC” (and not Cal BP) correct in the caption?

RESPONSE: Supplementary Table 2 has been updated to cal. yr. BP.

- Supplementary Fig. 6: In the caption, “[...]the Holocene-Pleistocene tradition” should be read as “[...]the Holocene-Pleistocene transition”.

RESPONSE: Supplementary Fig. 6 (now Supplementary Fig. 9) caption text has been corrected.

References

1. Adams, S. *et al.* Early human occupation of Australia’s eastern seaboard. *Sci. Rep.* **14**, 2579 (2024).
2. Clarkson, C. *et al.* Human occupation of northern Australia by 65,000 years ago. *Nature* **547**, 306–310 (2017).
3. Williams, A., Atkinson, F., Lau, M. & Toms, P. S. A glacial cryptic refuge in south-east Australia: human occupation and mobility from 36 000 years ago in the Sydney Basin, New South Wales. *J. Quat. Sci.* **29**, 735–748 (2014).

Reviewer #1:

I have now reviewed the revised ms (and the response to the original reviewers' comments). I think the authors have done an excellent job responding to most of the reviewers' comments. Please note that my point about the use of terms for major categories isn't entirely addressed in-text (e.g. referring to things to the "Pleistocene" isn't really useful, as that's such a broad span of time; much better to specify the particular time-frame at stake, e.g. refer to an envelope of time).

RESPONSE: the term Pleistocene has been replaced by "Late Pleistocene" and specified as (~35,000 – 11,700 years ago) as the time-frame of interest (see MS p. 2). The particular bracketing ages (~35,000-11,700) mark the envelope of time related to high-altitude mountain occupation in Australia and are established through reference to Supplementary Table 1. Throughout the manuscript Pleistocene has been changed to Late Pleistocene when it is not defined by reference to specific dates or layers (see MS and SI tracked changes).

In addition, Fig 1. - Pleistocene changed to 'Late Pleistocene' in key

I think there are still 2 other major (but easily addressed) and 3 minor (some stylistic) comments to address:

Major points:

Please explicitly tell the reader what the thickness of the stratigraphic interfaces are for each layer. For example, "the transition from Layer 1 to Layer is gradual, taking place over a depth of 2 cm". This is important because it tells the reader what the visible mixed zone between those layers are (interfaces, and their thicknesses, should always be commented on when reporting a stratigraphic sequence). Please note that this is not about the vertical distribution of artefacts (excellent to see these plotted on the section drawings; while it is standard to record artefact locations in 3 dimensional space in Australian archaeology, too rarely do researchers publish their locations on section drawings), but of the sediments themselves. For example, what is the thickness of the interface between Layers 9 and 10 (and why have some layers been distinguished by black lines on Figure 4, but not others? Again on Figure 4, why is there a subscript 4 after the median age of 5525 ? (it would be much easier to read the ages if from 10,000 onwards the thousands were separated by commas throughout the paper including all the figures and tables).

RESPONSE: The following text has been added to the MS, Archaeological Excavation methods section, p.9: There was a distinct boundary between the lowest Holocene Layer (8) and uppermost Late Pleistocene layers (Layers 9 and 10), and between the top of the in-situ deposit (Layer 3) and the overlying sand distributed during the 1980s (Layer 2). The clear distinction between these layers allowed them to be excavated as single contexts. To ensure no contamination between Layers 8 and 9/10, a thin (1cm) clean up unit was excavated after the excavation of Layer 8 and before the commencement of Layer 9. This yielded 3 stone artefacts, which were excluded from the analysis. The excavation of Layer 2 included the removal the contact interface to unsure Layer 3 did not contain any mixed material. There were no discernible interfaces within epochs, but

rather a gradual sedimentary build up over time, as evidenced in the sand grain size distribution (see Supplementary Figure 7) and dates (see Figure 3).

FIG 4. – why have some layers been distinguished by black lines, but not others?

RESPONSE: layers 5, 9 and 11 are not represented in this portion of the stratigraphic profile, and so were not distinguished by black lines. To address this concern, a dashed line has been added to Fig. 4 to indicate the relative depths of these layers. All layers are then distinguished by black lines.

FIG 4. - why is there a subscript 4 after the median age of 5525 ?

RESPONSE: this is a typo and has been deleted

It would be much easier to read the ages if from 10,000 onwards the thousands were separated by commas throughout the paper including all the figures and tables.

RESPONSE: A comma has been inserted throughout the manuscript and SI (see tracked changes) including all relevant figures (Fig 4 , Sup. Figs 2,4,5) and tables (Sup. Tables 1,2,3) to separate 10,000.

Please plot on the section drawings the excavation units. This is not about relating the 3D-plotted artefacts onto the section drawings, but additional to this, as it's very important for the reader to see how the excavation levels map onto (or don't) the stratigraphy. There will be times when an excavation unit cuts across a stratigraphic interface, for example, so while it's common to excavate stratigraphic interfaces separately (as their own excavation unit, to make sure there's no contamination of the materials included in a particular excavated stratigraphic layer), this doesn't always happen (and often stratigraphic interfaces can only be seen clearly AFTER an excavation is finished). The reader needs to be able to see how (1) excavation units, (2) stratigraphic layers, (3) in situ finds (the artefact dots presented in this paper), and (4) the carbon dates all are in relation to each other, on the section drawing. This is particularly important here given the paucity of stone artefacts in the lower levels where they occur. (I assume excavation units exist given that they refer to them e.g. in the caption of Supplementary Figure 4). If, on the other hand, the excavation did not subdivide layers into thinner excavation units, then please explicitly tell the reader so, and that stratigraphic interfaces were combined with either the overlying or underlying stratigraphic layer proper (in which case the chronostratigraphy cannot be further assessed by looking at the vertical pattern(s) and contexts of the excavated layers, sediments and their finds).

RESPONSE: the excavation units have been plotted on to the section drawings (see Sup. Figs 2,3,4).

Minor points:

I suggest using "cal BP" or "cal. BP" throughout, not "cal. yrs BP". The "yrs" is superfluous, and "cal BP" or "cal. BP" is more consistent with standard archaeological conventions.

RESPONSE: "cal. yrs BP" changed to "cal. BP" throughout (see MS and SI tracked changes).

Fig. 4 – image text changed from CalBP to cal. BP

Sup. Fig. 6 - image text changed from CalBP to cal. BP

Please standardise to "Dargan Shelter" throughout; at least one other time I read "Dargan rock shelter".

RESPONSE: *"Dargan Shelter" has been standardised throughout (see MS tracked changes, no changes required to SI).*

Note that Late Pleistocene, Early Holocene, Mid-Holocene and Late Holocene should all be capitalised, as they have all been ratified by the International Commission on Stratigraphy (see <https://stratigraphy.org/ICSchart/ChronostratChart2023-09.pdf>).

RESPONSE: *Late Pleistocene, Early Holocene, Mid-Holocene and Late Holocene have all been capitalised (see MS and SI tracked changes)*

Once these issues are addressed, I recommend the paper be published; it presents exciting archaeological findings, and is of both national and international, academic and popular interest.

Reviewer #2:

The authors have sufficiently addressed most of my comments and requests, and together with the changes implemented as a result of the other reviewer's comments, the manuscript has been significantly improved. I still think that the data supporting the human presence around 20.000 cal BP could be stronger, as only 2 lithic pieces belong to level 14 (plus another 2 in level 15), the hearth recorded in such layer is not so evident according to the provided photo, and only 1 radiocarbon date is available for that layer. Furthermore, the 4 mentioned lithic pieces are not identified among those photographed in Supplementary fig. 8 and I think that, at least those recorded in level 14, should be shown either by photos or drawings.

RESPONSE: *Supp. Fig. 8 – H17 shows the lowest complete flake, and is referenced in the MS, p. 6 "The lowest complete flake was found in layer 15. It was made from local claystone, showed use-wear on the distal end, measured 47 mm in length and weighed 6.9 g (see Supplementary Fig. 8 H-17)".*

Anyhow, even if these points cannot be addressed (as they were not in the last submission), I think the research deserves publication in a high quality scientific journal.